# Probenecid, an Old Drug with Potential New Uses for Central Nervous System Disorders and Neuroinflammation

**DOI:** 10.3390/biomedicines11061516

**Published:** 2023-05-24

**Authors:** Claudia García-Rodríguez, Paula Mujica, Javiera Illanes-González, Araceli López, Camilo Vargas, Juan C. Sáez, Arlek González-Jamett, Álvaro O. Ardiles

**Affiliations:** 1Centro Interdisciplinario de Neurociencia de Valparaíso, Facultad de Ciencias, Universidad de Valparaíso, Valparaíso 2360102, Chile; claudia.g.r@hotmail.com (C.G.-R.); paula.mujica@cinv.cl (P.M.); javi.illago@gmail.com (J.I.-G.); juancarlos.saez@uv.cl (J.C.S.); arlek.gonzalez@uv.cl (A.G.-J.); 2Escuela de Química y Farmacia, Facultad de Farmacia, Universidad de Valparaíso, Valparaíso 2360102, Chile; araceli.lopez@alumnos.uv.cl (A.L.); camilo.vargas@alumnos.uv.cl (C.V.); 3Instituto de Neurociencia, Facultad de Ciencias, Universidad de Valparaíso, Valparaíso 2360102, Chile; 4Escuela de Medicina, Facultad de Medicina, Universidad de Valparaíso, Valparaíso 2341386, Chile; 5Centro Interdisciplinario de Estudios en Salud, Facultad de Medicina, Universidad de Valparaíso, Viña del Mar 2540064, Chile

**Keywords:** probenecid, OAT, pannexin 1, TRPV2, Central Nervous System, neuroinflammation, epilepsy, neurodegenerative diseases

## Abstract

Probenecid is an old uricosuric agent used in clinics to treat gout and reduce the renal excretion of antibiotics. In recent years, probenecid has gained attention due to its ability to interact with membrane proteins such as TRPV2 channels, organic anion transporters, and pannexin 1 hemichannels, which suggests new potential therapeutic utilities in medicine. Some current functions of probenecid include their use as an adjuvant to increase the bioavailability of several drugs in the Central Nervous System (CNS). Numerous studies also suggest that this drug has important neuroprotective, antiepileptic, and anti-inflammatory properties, as evidenced by their effect against neurological and neurodegenerative diseases. In these studies, the use of probenecid as a Panx1 hemichannel blocker to reduce neuroinflammation is highlighted since neuroinflammation is a major trigger for diverse CNS alterations. Although the clinical use of probenecid has declined over the years, advances in its use in preclinical research indicate that it may be useful to improve conventional therapies in the psychiatric field where the drugs used have a low bioavailability, either because of a deficient passage through the blood–brain barrier or a high efflux from the CNS or also a high urinary clearance. This review summarizes the history, pharmacological properties, and recent research uses of probenecid and discusses its future projections as a potential pharmacological strategy to intervene in neurodegeneration as an outcome of neuroinflammation.

## 1. Introduction

Probenecid (PBN) is a fat-soluble derivative of benzoic acid, synthesized by Miller, Ziegler, and Sprague in 1949 [1]. Since its introduction to the market as Benemid in 1950 [1], it has been used for several applications in different pharmacological therapies. Still, for several reasons, such as the development of new-generation antibiotics, the use of PBN in clinics has decreased over time [2]. Initially, it was synthesized as an adjuvant to penicillin since PBN reduce its renal excretion and thus increase the antibiotic concentration in plasma and consequently prolongs its activity [3]. However, the use of PBN was limited by new antibiotics with better bioavailability and new antibiotic associations with a better efficacy [4]. Subsequent studies led to the fortuitous finding that PBN improved renal function by inhibiting the tubular reabsorption of uric acid, which showed its new therapeutic utility; thus, it began to be used for the treatment of gout [5,6]. However, it was later indicated that the decrease in serum uric acid and reduction in disease symptomatology correlated modestly compared to other uricosurics, and better therapies for the disease were found [7].

Between the 1950s and 1960s, some studies indicated that PBN blocks the active transport across the blood–brain barrier (BBB). It decreases the excretion of the acidic metabolites of dopamine (homovanillic acid) and serotonin (5-hydroxy indoleacetic acid), which led to several investigations in the psychiatric field [7]. Similarly, the coadministration of PBN with other compounds, such as morphine and fluorescein, was reported to increase the concentration of both agents in the brain and cerebrospinal fluid, whereas for hydroxyurea, coadministration with PBN did not affect its brain concentration [8,9,10]. In general, by knowing how PBN affects the transport of these drugs through the blood–brain barrier (BBB), it would be possible to use PBN in a beneficial way to increase the concentration of drugs in the brain. Regarding the transport of morphine through the BBB, it has been shown that it is a substrate of probenecid-sensitive transporters in the BBB. However, PBN-sensitive transport is not the most important mechanism of exclusion of morphine from the brain [10], which suggests that the use of PBN could help to reach a higher concentration of morphine in the brain without toxic risks as it is not the most important morphine exclusion pathway. Undoubtedly, more research is required to determine what happens with other potentially toxic agents, but it must be considered that PBN only acts on some of the BBB transporters, so there could be others that continue to work, which prevents the accumulation of possibly toxic substances in the brain.

In the field of basic research, PBN has gained significant relevance due to its reported ability to modify the activity of some membrane channels and transporters, including the blockade of pannexin-1 (Panx1) hemichannels and organic anion transporters (OAT) and the activation of TRPV2 channels [11,12,13]. In this review, we describe the use of PBN in various pathologies affecting the CNS, emphasizing those that present neuroinflammation as a common pathophysiological mechanism, including ischemia, sepsis, epilepsy, Parkinson’s, and Alzheimer’s diseases.

## 2. Probenecid: Pharmacology and Pharmacokinetics

PBN is a white crystalline powder that is poorly soluble in water but soluble in alcohol, chloroform, and acetone, and is also highly soluble in lipids. It has a pKa of 3.7, meaning it is a weak acid (Figure 1) [4]. It can be administered orally or intravenously. After oral administration, it is completely absorbed by the intestinal tract with a maximum administration of 3 g per day [14]. PBN has a high percentage of binding to plasma proteins since it binds 85–95% to albumin and has a volume of distribution of 0.003–0.014 L/kg. Depending on the dose, this drug has a half-life of 2 to 12 h. PBN diffuses freely across the BBB due to its high lipid solubility and is actively transported out of the brain [4,14]. PBN is metabolized in the liver and undergoes biotransformation by phase 1 metabolism, whereby the alkyl side chains are oxidized (about 70%), and phase 2 metabolism, whereby the conjugation reaction with glucuronic acid occurs (about 20%). Both products correspond to their major metabolites and, like PBN, block the tubular secretion of organic acids but have a lower binding affinity to plasma proteins. Therefore, they are excreted faster than PBN [4,14]. Byproducts of this drug are eliminated through the urine and correspond to 75–85% of the total PBN; a low percentage is excreted unchanged. Excretion is favored when the urine has a more alkaline pH [4,14].

PBN has several pharmacological interactions, including increasing the plasma concentration of some antibiotics, antivirals, antiretrovirals, nonsteroidal anti-inflammatory drugs, benzodiazepines, and loop diuretics [2,4,14]. It is assumed that this type of interaction is produced by PBN’s interference with the metabolism of these drugs or by its action on the blockade of drug transport, which induces an increased pharmacological effect through the accumulation of these agents in the CNS in the case of benzodiazepines [2]. Besides the interactions of PBN with antibiotics and their renal clearance, PBN also blocks the renal transport of some weak acid, basic, and neutral compounds [4]. There are no specific studies comparing the use of probenecid in the elderly and in those of other ages, but it must be considered that older people generally take more medication, so it is critical to become aware of whether PBN exhibit interactions with any of these drugs before recommending its use. In general, PBN has been considered to have a low toxicity profile [7]. The first large study of the side effects of PBN showed that it has little-to-no hemopoietic, renal, or hepatic toxicity [15]. The side effects of PBN include headache, nausea, vomiting, loss of appetite, gastrointestinal discomfort, and rash in 1 to 10% of people. These symptoms subside when the dose is readjusted [15]. In addition, it is contraindicated in patients with uric acid kidney stones, acute gouty arthritis, concomitant use with salicylates, and cases of hypersensitivity [2,4]. In recent decades, the effect of PBN on certain receptors and channels, such as the anion transporter blockade, TRPV2 channel activation, and Panx1 hemichannel blockade, has been studied (Table 1), conferring new potential therapeutic utilities [11,12,13].

## 3. Probenecid’s Pharmacological Targets

### 3.1. Organic Anion Transporter 1 and 3 (OAT1 and OAT3)

The organic anion transporter (OAT) subfamily is constituted of amphiphilic solute transporters that belong to the solute carrier 22 (SLC22) transporter family (Slc22a) [19]. OATs are efflux transporters involved in the outflow of organic anions and some drugs, toxins, hormones, and neurotransmitter metabolites (Figure 1). Within this family, OAT1 and OAT3 are expressed in the basolateral membrane of proximal renal tubule cells [19,20]. In the CNS, OAT1 is present in cortical and hippocampal neurons and ependymal and choroid plexus epithelial cells, whereas OAT3 is located on the luminal side of BBB and the apical side of the choroid plexus [19,20,21]. Since this drug inhibits these transporters [12,21], it is possible to increase the bioavailability of various compounds at the systemic and central levels, such as neurotransmitter metabolites and tryptophan metabolites (kynurenate) and diverse xenobiotics [22].

Pioneering studies indicated that PBN blocks the transport of neurotransmitter metabolites, including homovanillic acid (HVA, dopamine), vanillyl-mandelic acid (VMA, catecholamines), and 5-hydroxy indoleacetic acid (5-HIAA, serotonin) from the cerebrospinal fluid (CSF) to circulating blood, which limits their subsequent excretion in urine [23,24,25,26]. These findings lead to the use of PBN as a clinical test for studies of depression [27,28].

A further study reported that PBN administration increased the rat brain content of kynurenate, a metabolic product of tryptophan capable of antagonizing excitatory amino acid activity [29]. Other studies revealed that the peripheral coadministration of PBN and kynurenate increased brain and systemic levels of kynurenate by the blockade of OAT [30,31]. Accordingly, the peripheral inhibition of OAT3 by PBN reduces the elimination rate of the diuretic bumetanide, which increases its bioavailability in plasma; concomitantly, through OAT3 inhibition in the BBB, the transfer of bumetanide from the brain into the circulating blood is also reduced [12]. Another study provided a similar interpretation for the antioxidant N-acetyl cysteine. The increase in N-acetyl cysteine in plasma and brain tissue after the administration of PBN was interpreted as being due to the same mechanism of inhibition of OAT in renal tubules and the BBB [21]. The initial clinical use of PBN as a uricosuric agent was due to its inhibitory effect over OAT since it precludes the reabsorption of uric acid when binding to OAT in the proximal tubule, and this action favors the renal excretion of this molecule [14]. Nowadays, PBN is used in neuroscience research to prevent the leak of intracellularly loaded anionic fluorophores such as pH and calcium indicators to prevent the uptake of fluorescent dyes and to increase the bioavailability of other drugs, which makes it a useful tool to study the effect of different drugs at cellular and tissue levels.

### 3.2. TRPV2 Channel

The transient receptor potential V2 (TRPV2) is an ion channel belonging to the TRP-type channel family that is present in yeast, invertebrates, and vertebrates [32]. TRPV2 is an N-glycosylated protein composed of six transmembrane domains that form a channel with a central pore through which Ca^2+^ ions are transported into the cell (Figure 1). Their translocation to the plasma membrane is mediated by phosphatidylinositol 3-kinase (PI3K). As for the functions with which the TRPV2 channel is associated, it is relevant to mention cytokine secretion and release, chemotaxis, degranulation, osmotic balance, and somatosensitivity [33]. TRPV2 channels are expressed in sensory neurons from dorsal root ganglia; endothelial cells of the BBB; astrocytes; and striatal, hippocampal, and hypothalamic neurons [33,34,35,36,37]. The activity of this channel is initiated by physical stimuli, such as heat (>52 °C), cellular inflammation, and chemical stimuli, including cannabinoid ligands, 2-aminoethoxy diphenyl borate (2-APB), and PBN, among others [38]. This channel is of great interest due to its role in the proliferation and resistance of tumor cells to apoptosis. In fact, TRPV2 reduces glioma cell proliferation and promotes glial differentiation [39,40,41]. So, it has been proposed as a possible biomarker of various types of cancer, including brain tumors [42,43]. On the other hand, TRPV2 channel activation has been reported to contribute to proliferation, migration, tubulogenesis, and transendothelial electrical resistance (TEER) in brain endothelial cells [44], which highlights a potential role in the BBB. TRPV2 also plays an important role in early phagocytosis and innate immunity [45].

PBN specifically activates TRPV2 in mice over the other TRP-like receptors, as has been revealed by using calcium imaging techniques and electrophysiological recordings in HEK293T cells expressing the receptor [11]. In that study, 100 mM PBN had the best results regarding the concentration–response relationship of the TRPV2 channel. It was concluded that PBN elicits nociceptive-type reactions in mice, inferred by the selective action of PBN as an agonist and the functions performed by the TRPV2 channel [11]. Therefore, the study of TRPV2 is an area of research in which there is still a long way to go before determining its role in processes such as nociception and neuroinflammation, among others. Since PBN activates this channel, it could be beneficial to study the role of TRPV2 in those processes. Still, the therapeutic effect of PBN, which has been seen by reducing neuroinflammation, would not be related to this channel since TRPV2 activation could increase the influx of Ca^2+^, which could be detrimental to the cell. Indeed, PBN has been reported to induce mechanical hyperalgesia and allodynia by increasing the synaptic transmission in the rat dorsal horn [46]. Petitjean et al. reported that 500 μM of PBN induced calcium signals in isolated dorsal root ganglion neurons, which OAT and Panx1 hemichannel blockers did not prevent [46].

### 3.3. Pannexin-1 Channels

Panxs were originally described as proteins homologous to innexins found in invertebrates [47]. They are channel-forming proteins with a relevant role in autocrine and paracrine cellular communication [48]. The oligomerization of seven subunits of these proteins forms a functional hemichannel in the plasma membrane [49,50], which allows for the flow of small substances, including signaling molecules (i.e., ATP and glutamate), substrates (glucose), ions (chloride), and positively (ethidium and DAPI) and negatively (calcein and Lucifer yellow) charged dye molecules [51].

The genes composing this family are *PANX1*, *PANX2*, and *PANX3*. The first two genes are highly expressed in the CNS and are abundantly expressed in the cerebellum, thalamus, cerebral cortex, and hippocampus. It is specifically expressed in neurons, interneurons, astrocytes, and microglia [52,53,54,55,56,57,58]. The Panx1 hemichannels exhibits a postsynaptic distribution colocalizing in neurons with the synaptic machinery, and their activation negatively controls glutamatergic synaptic transmission and plasticity [58,59,60,61]. In contrast, Panx1 is upregulated under inflammatory conditions in microglia and astrocytes, which suggests that Panx1 hemichannels contribute to inflammatory responses in the case of injury or infection [48].

Panx1 hemichannels exhibit two modes of activity: constitutive small-pore ion channel activity characterized by the low conductance (50–80 pS) of slightly selective anion permeation (chloride ions) and small molecules such as DAPI and ATP [62,63,64,65,66], and large pore activity with high conductance (100–550 pS) mediating a nonselective ionic flux [67,68,69] responsible for ATP release and the permeation of positively (i.e., ethidium bromide, DAPI, and others) and negatively (i.e., calcein and Lucifer yellow) charged molecules [13,54,67,69,70,71]. Panx1 hemichannels are activated by intense neuronal activity [72,73]; high concentrations of external potassium ions [74]; the activation of CaMK II [66]; low oxygen conditions [67,71]; interactions with NMDAR [72,75], P2X7R [70], α1AR [76], and α7-nAChR [77]; and the caspase-dependent cleavage of its carboxy terminal [78].

Silverman et al., using electrophysiological techniques, showed that PBN abolished Panx1 hemichannel currents at a 1mM concentration in oocytes expressing Panx1 in a concentration-dependent fashion [13]. The IC_50_ of PBN observed was 150 μM in oocytes. However, in the human cell line (HEK293), PBN inhibited Panx1 hemichannel currents with an IC_50_ 350 μM without altering purinergic signaling (P2X7R) [79]. Interestingly, the inhibition of Panx1 hemichannels has been linked to various processes occurring in the brain, such as neuroinflammation and neuronal plasticity [48,51]. Neuroinflammation is a common secondary factor of different pathologies, including neurodegenerative diseases and cerebrovascular injury [48,51]. Activation of the inflammasome is crucial in the inflammatory process, so blocking Panx1 hemichannels with PBN has been suggested to decrease inflammation [47]. It is important to remember that Panx1 also forms functional gap junction channels between cells, which are insensitive to known Panx1 hemichannel blockers, including PBN and carbenoxolone [80].

### 3.4. Other Targets

#### 3.4.1. Purinergic Receptors

Purinergic receptors are a family of plasma membrane proteins classified into two categories, P1 or adenosine receptors (ARs) and P2 receptors [81]. The P2 receptors are subgrouped into ligand-gated ion channels (P2X) and G protein-coupled metabotropic receptors (P2Y) [81]. Currently, different roles are attributed to ATP signaling in the CNS through the activation of purinergic receptors expressed in brain cells: from cell communication, migration, proliferation, and growth upon physiological conditions to neuroinflammation, brain injury, chemotaxis to the injured site, cell stress, and neurodegeneration under pathological conditions [81,82].

Since Panx1 hemichannels can release ATP, which is inhibited by PBN [13], and considering that the purinergic signaling and receptors have been associated with Panx1 hemichannel activity [83,84], PBN could modulate the purinergic signaling by the decrease in extracellular ATP and the activation of purinergic receptors. However, it has been described that PBN blocks P2X7R via a Panx1 hemichannel-independent mechanism [18]. Bhaskaracharya et al. reported that PBN inhibits P2X7R-mediated dye uptake in transfected HEK-293 cells at an IC_50_ of 203 μM [18]. Furthermore, PBN also reduced interleukin (IL)-1β secretion from human CD14^+^ monocytes and P2X7R calcium currents in HEK-293 cells at a 1 mM concentration, which shows that PBN is involved in inflammation and directly blocks the human P2X7Rs [18].

#### 3.4.2. Organic Cation Transporters (OCTs)

Some evidence indicates that PBN can also decrease the renal clearance of organic cations [85,86,87], which suggests the inhibition of organic cation transporters. Like OAT, the organic cation transporter (OCT) is a subfamily of transporters belonging to the SLC22A family that facilitates the intracellular uptake of small organic cations, including amino acids, vitamins, neurotransmitters, and creatinine [22]. These transporters are widely distributed in mammalian tissues, and three OCT subtypes (OCT1, OCT2, and OCT3) have been identified [88]. OCT members are expressed in epithelial cells from the kidney and liver, where PBN was found to inhibit the uptake of organic cations such as cimetidine and famotidine at the renal level [85,87,89]. In fact, although organic cation transporters were thought to only have an affinity for cationic organic compounds, they have also been reported to have an affinity for compounds that inhibit anionic organic transporters, such as PBN. In this sense, 1 mM PBN has been reported to reduce cimetidine transport in rabbits in a dose-dependent manner [89]. In 1988, Hysy et al. described that PBN at a 10 mM concentration reduced N’-methylnicotinamide (NMN) transport through organic cation transporters in renal proximal tube brush-border membrane vesicles (BBMV) of rabbit [90]. It has been described that 500 mg of PBN administered orally inhibits the renal excretion of cimetidine in humans [85]. Similarly, it was observed that 1500 mg of PBN administered orally in humans had an inhibitory effect on famotidine renal excretion [87]. In the CNS, OCT2 and OCT3 are expressed in neurons and glial cells from several brain areas, choroid plexus, and ependymal cells, and their principal role is the transport of neurotransmitters such as serotonin, norepinephrine, dopamine, histamine, choline, coenzymes, drugs, and xenobiotics [88,91]. On the other hand, OCT1 is expressed in hippocampal neurons and choroid plexus epithelial cells in humans and rodents. At the same time, a lower expression is detected in both luminal and abluminal membranes from endothelial BBB cells, which contributes to the transport of serotonin, monoamines, acetylcholine, and histamine (reviewed in [91]). However, the direct effect that PBN’s selective inhibition has on OCTs in the brain has not been well studied; rather, studies have focused on the renal system. Nevertheless, it has been proposed that high levels or the long-term use of cimetidine increases serum levels of prolactin (Hyperprolactinemia) and hence increases the synthesis and secretion of dopamine [92], so PBN, if co-administered with cimetidine, could indirectly impact the dopaminergic system. In this regard, Werdinius began to publish studies on the effect of PBN on monoamine levels in the brain [93]. Subsequently, several studies were carried out in rodents where an influence of PBN on the availability of monoamines was described [94,95]. The field of OCTs remains to be explored when it comes to the influence of PBN on neurotransmitter levels.

## 4. Central Nervous System Diseases and the Role of Probenecid

### 4.1. Neuroinflammation

Neuroinflammation refers to the inflammatory response within the CNS caused by various pathological factors or insults, including infection, trauma, ischemia, misfolded proteins, and toxins [96]. Inflammasomes are multimeric protein complexes in the cytosol of all cells, including stimulated immune cells that control the inflammatory response [97]. Inflammasomes activate the proinflammatory caspase-1 after their activation, which cleaves the propeptides pro-IL-1β and pro-IL-18 into the mature cytokines IL-1β and IL-18, which are then secreted by the cells. Caspase-1 can also induce pyroptosis, a proinflammatory form of cell death, which releases proinflammatory signals [98]. The inflammasomes can be classified by the stimuli that activate them. In the CNS, the inflammasome comprises the nucleotide-binding oligomerization domain, leucine-rich repeat-containing family proteins (NLRP), adaptor protein ASC, and caspase 1 enzyme [97]. Diverse stimuli, including toxins derived from viruses or bacteria, misfolded proteins (i.e., amyloid beta-peptide and alpha-synuclein), reactive oxygen species (ROS), and elevated extracellular ATP and K^+^ concentrations activate NLRP. The latter acts as membrane receptors that sense and bind these signals to recruit and promote inflammasome complex assembly [97,99]. Other stimuli, such as cytosolic double-stranded DNA (dsDNA), can activate the AIM2-like receptors (ALRs) [100], and pyrin receptors respond to toxins that covalently inactivate the small GTPase RhoA [101]. These receptors also activate and shape the different inflammasomes [98]. Once the inflammasome is activated, it can exert its proinflammatory function through the activation of caspase-1, the release of cytokines IL-1β and IL-18, leading to pyroptosis.

The outflow of ATP and the consequent activation of P2X7Rs also promotes inflammasome activation (NLRP3, ACS protein, and caspase-1 activation) [47], which leads to IL-1β production [97]. IL-1β, released into the extracellular space, reduces intercellular communication through gap junctions while increasing connexin 43 hemichannel activity in astrocytes [102]. The role of Panx1 hemichannels on the inflammasome was first described by Silverman et al., who studied caspase-1 activation using astrocytes and neurons expressing or lacking Panx1. In wild-type cells, normal caspase activation was induced by stimulation with elevated extracellular K^+^, whereas in Panx1-knockout cells, caspase activation was undetectable, which supports the idea that Panx1 hemichannels mediate inflammation [74]. Moreover, in that study, preincubation with 1 mM PBN completely prevented caspase-1 activation in neurons and astrocytes, which further confirmed that PBN can alleviate inflammation and showed that PBN is a new tool that can reduce cell death in a variety of CNS lesions and diseases [74].

Panx1 participates in ATP-induced ATP release, and the mechanism involves purinergic receptors (P2YRs) that are coupled with G-protein and activate phospholipase C and IP_3_ production. High levels of extracellular ATP induce a rapid outflow of K^+^ from the innate immune cells [74,103,104] and Ca^2+^ influx through purinergic receptors (i.e., P2X7Rs) that interact with Panx1 hemichannels [105]. Therefore, during tissue damage, an increased Panx1 hemichannel activity promoting ATP release acts as a central damage-associated molecular pattern (DAMP) [106,107]. Notably, in a neuron–astrocyte coculture system, the ATP and glutamate released from astrocytes treated with conditioned media from inflammatory microglia promote neuronal death by neuronal Panx1 hemichannel activation [108,109]. In addition, increased extracellular IL-1β functions as a signal to recruit more immune cells, which enhances neurotoxicity. In this sense, the Panx1 hemichannel plays a fundamental role in the initial events of the signaling and activation of the inflammasome [106,110], a multiprotein complex that mediates interleukin (IL-1β and IL-18) production and secretion [97]. This inflammatory environment negatively affects CNS cells as they deprive neurons of the astroglia’s protective spatial buffering function, which increases neuronal vulnerability and the incidence of neuronal death [108].

Regarding the inhibition of Panx1 hemichannels with PBN, it has been described that the activation of the inflammasome has been evaluated in cultured astrocytes after oxygen–glucose deprivation (OGD) using different PBN concentrations. In that study, the authors concluded that the protein expression levels of aquaporin-4 (AQP4), NLRP3, and caspase-1 increased concomitantly after 6 h of OGD. This increase was strongly inhibited by PBN treatment applied before OGD [47]. Additionally, PBN significantly improved astrocyte survival, and this was associated with reduced ROS production and NLRP3, caspase-1, and IL-1β expression. Recently, Zheng et al. described the protective effect of PBN by blocking Panx1 hemichannels in brain lesions by inhibiting the AIM2 neuronal inflammasome after a subarachnoid hemorrhage in rats. The PBN was administered orally (1 mg/mL in water, 50 mL final volume) and through intraperitoneal injections (1 mg/kg) twice, before and after 2 h of hemorrhage. PBN decreased the levels of the AIM-2 inflammasome, ASC protein, caspase 1, P2X7R, IL-18, IL1β, purinergic receptors, and reactive oxygen species (ROS) [111].

According to the above findings, inhibition at the early stages of the inflammatory process with PBN could be crucial to stop neuroinflammation and neuronal death, which is a common hallmark of several pathologies; thus, this opens the possibility of new uses of PBN.

### 4.2. Epilepsy

Epilepsy is one of the most common neurological disorders that affects brain function. It is characterized by spontaneous seizures leading to cognitive, neurological, and psychosocial consequences [112]. Epilepsy affects around 50 million people worldwide and is responsible for a 2–10% reduction in life expectancy [113]. Additionally, approximately one third of epileptic patients do not respond to any of the currently available antiepileptic drugs, and for them, the only option is brain surgery, if possible [114]. For this same reason, it is of great interest to be able to develop new therapies against this disease with a high prevalence in the world. PBN could be of interest in the pharmacology of epilepsy since Panx1 hemichannels are involved in the pathophysiology of epilepsy [115]. Panx1 contributes to the generation and prolongation of epileptic seizures [116,117]. In fact, Dossi et al. reported that the Panx1 hemichannel blockade with 1 mM of PBN inhibited the induction of ictal discharges (IDs) and decreased seizure frequency in cortical slices obtained from postoperative epileptogenic tissues of patients with epilepsy [118]. In the same study, a single 200 mg/kg PBN intraperitoneal injection decreased the frequency of spontaneous seizures in mice with temporal lobe epilepsy (TLE) treated with kainic acid. In another study, PBN was reported to decrease the onset and severity of seizures [119]. In mice injected with 80 mg/kg pentylenetetrazol (PTZ), a chemical kindling model of epilepsy, Aquilino et al. demonstrated that pretreatment with PBN reduced the severity of seizures and the time to reach stage 5 seizures on the Racine scale, while it increased the survival after the PTZ injection [119]. It is important to note the implication of the connexin gap junctions in epilepsy. The glial proliferation and reduced gap junctional communication that occurs in epilepsy can have detrimental consequences [120], which can be prevented or reversed by the connexin gap junction blocker carbenoxolone [121,122] that would further reduce the gap junctional communication worsening the cell–cell coupling scenario. Alternatively, the antiepileptic effect of carbenoxolone could be explained by its inhibitory effect on Panx1 hemichannels since there is evidence that carbenoxolone also blocks these hemichannels [123]. All this evidence supports the potential role of PBN as an antiepileptic agent (Table 2).

### 4.3. Parkinson’s Disease

Parkinson´s disease (PD) is the second most common neurodegenerative disorder affecting 2–3% of the population over 65 years of age; it is characterized by the deterioration of motor activities due to the progressive loss of dopaminergic neurons in the substantia nigra [135]. This condition causes a striatal dopamine deficiency, intracellular inclusions of α-synuclein, and the formation of Lewy bodies, which are the neuropathological hallmarks of PD [136]. PD has several stages, from mild tremors to absolute dependence [135]. Most PD cases are sporadic and associated with aging, whereas 5–10% of PD cases are genetic and associated with mutations in PARK genes, which encode proteins such as α-synuclein, LRRK2, parkin, PINK, and DJ-1 [135,136]. Furthermore, nongenetic risk factors include environmental toxins, pesticides, heavy metals, traumatic lesions, and bacterial or viral infections, which are closely associated with the inflammation that promotes the manifestation of Parkinson´s disease [137]. In this regard, Panx1- and connexin-based channels and inflammation have been proposed to be involved in PD [138]. Indeed, α-synuclein was found to induce the opening of Panx1 and connexin 43 hemichannels, and the intervention of these channels was suggested as a potential therapeutic target in α-synuclein-associated diseases [139].

On the other hand, the effects of the intraperitoneal administration of L-kynurenine (kynurenic acid precursor), a competitive antagonist of the glutamate receptor type (NMDAR), together with PBN was evaluated in a 6-hydroxydopamine (6-OHDA)-induced PD mice model. In that study, the coadministration of L-kynurenine and PBN at doses of 75 mg/kg and 50 mg/kg, respectively, for 7 days reduced the decay in the total dopamine levels and protected mice from striatal damage and neurodegeneration [131]. Silva-Adaya et al. concluded that the protective effects are due to the product of kynurenic acid, and the function of PBN was to increase kynurenic acid levels in the brain [131].

As mentioned above, PBN has also been used to increase 1-methyl-4-phenyl-1,2,3,6-tetrahydropyridine (MPTP) levels in the brain, and this can be used to generate a mouse model for PD. In that model, two studies were conducted. The first study examined the benefits of methylene blue when administered in an MPTP/PBN model mouse [124], and the second study evaluated the role of Toll-like receptors 4 (TLR4) in PD [130] (Table 1). The use of PBN as an alternative therapy for PD is not entirely clear, even though the disease is related to changes in Panx1 hemichannel activity. In that context, the use of PBN in these studies was aimed at increasing the bioavailability of other molecules in the CNS, either as a therapeutic agent for PD or as an adjuvant for the induction of the pathogenic mechanism in animal models of PD.

### 4.4. Alzheimer’s Disease (AD)

Among the neurological conditions affecting older adults, AD is the most prevalent chronic neurodegenerative disease and the most common cause of dementia worldwide. AD is characterized by brain atrophy and the accumulation of amyloid plaques and neurofibrillary tangles that constitute the hallmarks of the disease [140]. AD comprises several stages starting with recent memory loss and progressively advancing to affect other cognitive domains [141]. One of the earliest events that correlate with cognitive impairment is synaptic loss [142,143] due to the accumulation of soluble amyloid oligomers, which freely diffuse and bind to proteins in the brain [144]. Amyloid oligomers are aggregates of amyloid beta-peptide (Aβ), a fragment of 40–42 amino acids with great facility to self-aggregate generated from the proteolytic cleavage of amyloid precursor protein (APP) [145]. These aggregates subsequently form amyloid fibrils that constitute insoluble extracellular deposits called amyloid plaques [140]. Together with amyloid plaques, neurofibrillary tangles, formed by hyperphosphorylated Tau protein aggregates, constitute the brain lesions that appear in the brain structures that are important for cognitive function [146].

In that pathological context, the soluble Aβ oligomers impair synaptic plasticity by reducing the size and number of dendritic spines, which prevents long-term potentiation (LTP), promotes long-term depression (LTD), and leads to cognitive dysfunctions and an impaired spatial memory [144]. Interestingly, and as mentioned before, it has recently been proposed that Panx1 hemichannels could play an important role in that disease since they modulate the induction of excitatory synaptic plasticity and paracrine communication [51], which are two processes that are affected during the development of this disease. As a matter of fact, a study evaluated the in vitro effect of PBN in a mouse model of Alzheimer’s disease (transgenic mice carrying mutations in the APP and PSEN 1 genes), where it was observed that the acute blocking of Panx1 hemichannels with 100 µM PBN normalizes hippocampal synaptic plasticity and improves the morphology and density of dendritic spines [126]. Flores-Muñoz et al. also observed a reduction in the levels of the active form of the p38 MAPK, which is increased in the Alzheimer’s brain and is associated with the neurotoxic phase in early AD [147] (Hensley et al., 1999). However, the neurodegeneration hallmarks were not reduced by the acute PBN treatment, likely due to the form of administration, namely the in vitro treatment of brain slices. Thus, the direct effect of PBN in AD is still preliminary, and additional studies using prolonged forms of administration require further research. In another study, rats were treated with an intrathecal injection of amyloid-beta peptide that was coadministered with PBN and L-kynurenine (50 mg/kg/7 days and 75 mg/kg/7 days, respectively) to evaluate its effect on Aβ-induced neurotoxicity. Carrillo-Mora et al. described that those injected rats experienced an improved spatial memory and decreased reactive gliosis in the CA1 region of their hippocampi [125]. It should be noted that the study was focused on demonstrating the positive effects of kynurenic acid and its NMDAR antagonism in rats. Therefore, the protective actions were not exclusively linked to the use of PBN and were administered to interrupt the excretion of kynurenic acid from the CNS, and the objectives were similar to those of the studies mentioned for PD [131]. In the same line, various molecules including sodium azide, 2,4-dinitrophenol, verapamil, nifedipine, quinidine, MK-571 (leukotriene D4 receptor antagonist), and PBN were used to evaluate the passage through the BBB and brain availability of cytosine (CTS), an experimental treatment for AD. Yu et al. reported that the oral administration of 200 μM PBN increased the uptake and decreased the outflow of CTS from the rat brain through endothelial cells [133].

The use of PBN as a therapeutic agent for AD has still not been entirely explored since, in most of the exposed cases, it was used to increase the bioavailability in the CNS of other molecules, and in the case where it was administered as a mono drug for AD treatment, it did not achieve sufficient benefits, and the evidence is still preliminary.

### 4.5. Other Diseases

PBN has also been used in several animal and cell models of CNS disorders. For instance, PBN was shown to have a protective effect in a Huntington’s disease (HD) model, a progressive and autosomal dominant neurodegenerative disorder caused by an expanded CAG repeat in the huntingtin gene which encodes an abnormally long polyglutamine repeat in the huntingtin protein. HD is characterized by movement disorders (including chorea and a loss of coordination) and cognitive decline defects [148]. In the study, Vamos et al., using the N171-82Q transgenic HD animal model, found that PBN administration improved motor behavior, increased cell survival, and significantly reduced neuronal loss and the number of intranuclear neuronal aggregates [149].

PBN was also used in an animal model of multiple sclerosis (MS), an experimental autoimmune encephalomyelitis (EAE) model [127,128]. MS is a progressive, autoimmune neurologic disorder of the CNS characterized by demyelination, axon degeneration, and neuroinflammation [150]. Autoreactive T helper cells (Th1 and Th17), which cross the BBB, are responsible for the inflammatory and demyelinating condition in MS. Hainz et al. showed that pretreatment with PBN prevents some early EAE symptoms, including motor abnormalities and diminished T cell numbers in the CNS and cell infiltrates in the spinal cord [127]. Moreover, in pronounced EAE after 20 days of treatment with PBN, inflammation, T cell infiltration, and oligodendrocyte cell loss were reduced, indicating that PBN prolonged neuronal and glia survival and prevented the progression of clinical symptoms associated with the EAE model of MS [128]. Interestingly, in that study, the effects of PBN were mediated by Panx1 hemichannel inhibition, confirming that PBN alleviates MS symptoms by reducing neuroinflammation.

In another study, Wei et al. evaluated the protective effect of the intravenous, intraperitoneal, and gavage administration of PBN against cerebral ischemia/reperfusion (I/R) injury in rats. Independent of the route of administration, PBN decreased neuronal death in the CA1 area, and some inflammatory markers were induced by I/R [132]. A similar study evaluated PBN treatment in a mouse model of sepsis-associated encephalopathy (SAE) induced by cecal ligation and puncture, which was characterized by cerebral dysfunction with varying neurological symptoms. In that study, Zhang et al. reported that PBN reduced the overexpression of inflammatory mediators such as the tumor necrosis factor-α (TNF-α), IL-6, and IL-1β in the hippocampus, which appear when acute cerebral dysfunction occurs [134]. Furthermore, cognitive impairment was also reduced after PBN administration. Notably, the reduced neuroinflammatory response and ATP release were mediated by a Panx1 hemichannel blockade, which suggests that PBN could be a promising therapeutic approach for treating inflammation during the cerebral dysfunction of sepsis.

On the other hand, PBN also has been used to alleviate migraine headaches [129]. Cortical spreading depression is thought to cause the migraine aura by activating perivascular trigeminal nerves [151]. In this report, Karatas et al. identified a signaling cascade involving the neuronal Panx1 hemichannel opening and caspase-1 activation followed by high-mobility group box 1 (HMGB1) release from neurons and nuclear factor κB activation in astrocytes [129]. Panx1 hemichannel blockers, including PBN, suppressed this inflammatory response and abolished the headaches, which further supports the relationship between Panx1 hemichannels and inflammation.

In addition to its use in the mentioned diseases, PBN has also been used in other studies to discriminate between the effect of connexin and Panx1 hemichannels since PBN only blocks Panx1 hemichannels and not the gap junction channels formed by Panx1 [80]. Thus, it was possible to determine that connexins, but not Panx1 hemichannels, have a very important role in the generation of endogenous spontaneous electrical activity in subplate neurons in the developing brain [152] (Figure 2).

## 5. Conclusions

This review summarizes the recent research uses of PBN, a drug formerly used for gout treatment, as a pharmacological agent for some CNS disorders. PBN uses are related to their identified targets, including the blockade of OATs and Panx1 hemichannels and the activation of TRPV2 channels. Among the identified functions of PBN are its use as an adjuvant to increase the bioavailability of several drugs in the CNS. More recent preclinical evidence suggests the neuroprotective, antiepileptic, and anti-inflammatory properties of PBN in animal models of neurological and neurodegenerative conditions. Among these studies, the use of PBN as a Panx1 hemichannel blocker that reduce neuroinflammation is highlighted, and the neuroinflammation is a major cause of diverse CNS alterations. In this regard, it is important to note that neuroinflammation is a broad and complex response that involves several molecular cascades activated by specific molecular events that result in the release of inflammatory mediators and the activation of cellular targets such as receptors, channels, and enzymes. Therefore, the intervention of specific pathways does not discard the recruitment of other routes. Likewise, potential differences between rodent and human inflammasome types and species-dependent activation mechanisms should be considered. In this regard, it is noteworthy that in many cases, the evidence obtained from animal studies can not necessarily be translated into human studies. Regardless, the preclinical evidence of an anti-inflammatory-like effect of PBN reviewed here paves the way for future studies. The evidence summarized here also highlights the ability of PBN to reduce epileptic discharges in studies in rats and the severity of convulsions upon the induction of status epilepticus.

Although its clinical use has declined over the years, advances in the use of PBN in the preclinical research suggest that it may be useful to improve conventional therapies in the psychiatric field where the drugs used have a low bioavailability, either because of their deficient passage through the BBB or a high efflux from the CNS, or also a high urinary elimination rate. Based on the information gathered, it is recommended to continue studying the drug PBN in neuroinflammatory diseases, and particularly its possible benefit at the hippocampal level in AD since positive results of its use were evidenced in these cases. In addition, their capability to cross the BBB and their ability to increase the bioavailability of several metabolites in the CNS support their use.

## Figures and Tables

**Figure 1 biomedicines-11-01516-f001:**
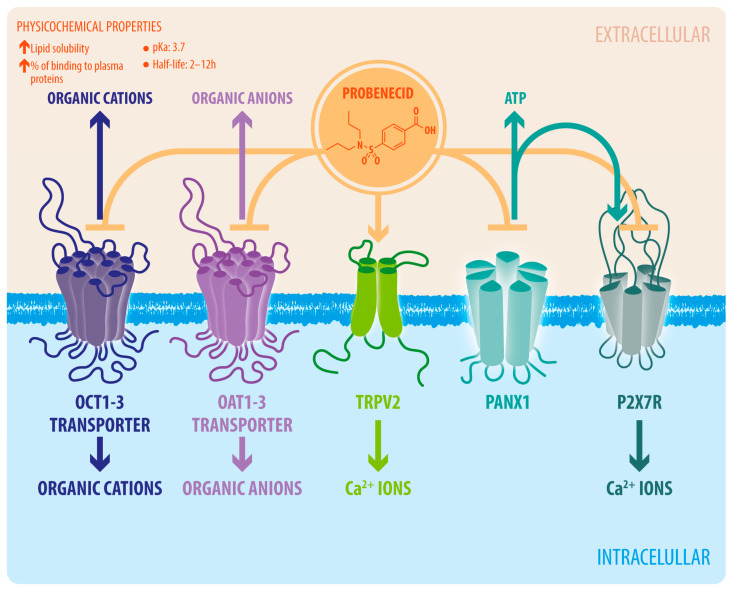
Structure and pharmacological brain targets of probenecid. In the brain, organic anion and cation transport are regulated by the family of organic anion transporter (OAT) isoforms 1 and 3 and organic cation transporter (OCT) isoforms 1, 2, and 3 present in the choroid plexus and brain–blood barrier. Probenecid blockade of OAT and OCT increases the levels of endogenous organic anions and cations in blood, cerebrospinal fluid, and the brain. TRPV2 channels mediate the influx of Ca^2+^ ions into the neurons and glial cells. Probenecid activates TRPV2 channels, which increases Ca^2+^ entry and Ca^2+^-dependent signaling. Panx1 hemichannels mediate the ATP release from neurons and glial cells to the extracellular space, which contributes to paracrine cell communication. Indirectly, Panx1 hemichannels can also mediate Ca^2+^ entry through the activation of P2X7 receptor (P2X7R) by ATP.

**Figure 2 biomedicines-11-01516-f002:**
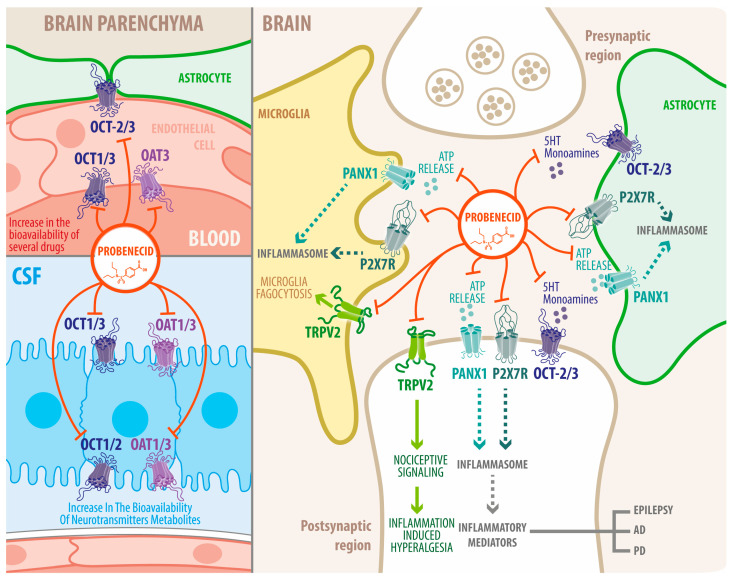
Potential functions of probenecid in brain tissue and their contribution to neurological diseases. In the brain, OAT isoforms 1 and 3 (OAT1–3) and OCT isoforms 1, 2, and 3 are present in the parenchyma (neurons and glial cells), ependymal cells, choroid plexus epithelial cells, and endothelial cells from the blood–brain barrier (BBB) where they regulate the transport of organic anions and cations from and to the cerebrospinal fluid (CSF) via the choroid plexus and from and to the brain via the BBB. Probenecid blockade of OAT and OCT increases xenobiotics levels, systemic drugs in blood and CSF, neurotransmitter metabolites, and endogenous molecules in CSF and brain parenchyma. TRPV2 channels are expressed in neurons and glial cells, mediating Ca^2+^ influx and Ca^2+^-dependent intracellular signaling. Probenecid activates TRPV2 channels to facilitate TRPV2 functions such as inflammatory and nociceptive signaling and microglial phagocytosis. TRPV2 is also expressed in brain endothelial cells, which could contribute to proliferation, migration, tubulogenesis, and transendothelial electrical resistance. Nevertheless, an excessive TRPV2 activation by probenecid could exacerbate the contribution to inflammation-induced hyperalgesia. Panx1 hemichannels are expressed in neurons and glial cells, facilitating paracrine communication through ATP release. ATP activates the P2X7Rs, promoting an increase in Ca^2+^ influx. Panx1 hemichannels and P2X7Rs are upregulated under inflammatory conditions, contributing to inflammasome activation and inflammatory response, although Panx1 blockade does not prevent Ca influx by P2X7R. Several neurological and neurodegenerative disorders are characterized by neuroinflammation, so probenecid, by blocking Panx1 hemichannels, would alleviate inflammatory conditions promoting cell survival.

**Table 1 biomedicines-11-01516-t001:** IC_50_ or EC_50_ values (μM) of probenecid.

Target	IC_50_/EC_50_	Assay	Reference
OAT1	IC_50_ 12.3 μM ^1^	[^14^C]PAH uptake	Takeda et al., 2001 [16]
OAT3	IC_50_ 4.93 μM ^1^	[^3^H]ES uptake	Takeda et al., 2001 [16]
TRPV2	EC_50_ 31.9 μM ^2^	Calcium currents	Bang et al., 2007 [11]
Panx1	IC_50_ 150 μM ^3^	Ionic currents	Silverman et al., 2008 [13]
OCT1	IC_50_ 1640 μM ^4^	[^14^C]TEA uptake	Arndt et al., 2001 [17]
OCT2	IC_50_ 1700 μM ^4^	[^14^C]TEA uptake	Arndt et al., 2001 [17]
P2X7R	IC_50_ 203 μM ^5^	Ethidium uptake	Bhaskaracharya et al., 2014 [18]

^1^ Human OAT1 and OAT3 transfected in S2 cells. ^2^ Rat TRPV2 transfected in HEK293 cells. ^3^ Human Panx1 transfected in *Xenopus* oocytes. ^4^ Rat OCT1 and OCT2 transfected in *Xenopus laevis* oocytes. ^5^ Human P2X7R transfected in HEK293 cells.

**Table 2 biomedicines-11-01516-t002:** Effects of PBN in Central Nervous System diseases.

Study	Type of Study	Study Model	Doses or Concentration	CNS Pathology	Probenecid Effect
Aquilino et al., 2020 [119]	In vivo	Mice pretreated with PBN are exposed to 80 mg/kg de PTZ	250 mg/kg i.p.	Epilepsy	Decrease in seizures severity.
Biju et al., 2018 [124]	In vivo, ex vivo	MPTP mice/PBN and behavioral assessment and tyrosine hydroxylase (TH) neuron analysis	250 mg/kg i.p.	PD	Low-dose methylene blue has neuroprotective actions in PD.
Carrillo-Mora et al., 2010 [125]	In vivo, ex vivo	Coadministration of kynurenic acid and PBN in beta-amyloid peptide rats. Evaluation by locomotor, memory, and morphological tests	50 mg/kg i.h. or i.p.	AD	Improvements in spatial memory and decrease in neurodegenerative events.
Dossi et al., 2018 [118]	Ex vivo, in vivo	Postoperative samples of human tissue in patients with epilepsy Mouse model with kainic acid of temporal lobe epilepsy (TLE)	1 mM 200 mg/kg i.p.	Epilepsy	Significant decrease in epileptic discharges.
Flores-Muñoz et al., 2020 [126]	Ex vivo	Transgenic APP/PS1 mice were dissected in different histological sections, to which PBN was administered	100 μM	EA	Decrease in synaptic plasticity deficits and improvement in dendritic spine density and dendritic arborization.
Hainz et al., 2016 [127]	In vivo	Experimental autoimmune encephalomyelitis (EAE) mouse model–multiple sclerosis (MS) mouse model	200 mg/kg i.p.	EAE/MS	Significant decrease in inflammation and infiltrating T cells in the CNS.
Hainz et al., 2017 [128]	In vivo	Experimental autoimmune encephalomyelitis (EAE) mouse model–multiple sclerosis (MS) mouse model	200 mg/kg i.p.	EAE/MS	Decrease in inflammation and T-cell infiltration and increase in oligodendrocyte number.
Jian et al., 2016 [47]	In vitro	Primary neuron and astrocyte culture from newborn mice exposed to oxygen–glucose deprivation/reoxygenation (OGD/RX)	5–10 μM	Ischemia	Inhibition of inflammasome and caspase 1 activities.
Karatas et al., 2013 [129]	In vivo	Experimental mice model of cortical spreading depression (CSD) induced by pinprick or KCl	60 μg i.c.v.	Migraine/headache	Suppression of trigeminovascular activation, dural mast cell degranulation, inflammation, and headache.
Shao et al., 2019 [130]	Ex vivo	MPTP mice/PBN and subsequent substantia nigra and striatum analysis	250 mg/kg i.p.	PD	Verification of the neuroprotective role of TLR4 in PD.
Silva-Adaya et al., 2011 [131]	In vivo	6-OHDA-induced PD model mice, coadministration of PBN with L-kineurin	70 mg/kg i.p.	PD	Increase in CNS kynurenic acid levels.
Silverman et al., 2009 [74]	In vitro	Primary neuron and astrocyte culture. Culture of oocytes absent from follicular cells of Xenopus laevis frogs	2 mM	NA	Blockade of inflammasome activation and PANX1 currents.
Sun et al., 2001 [8]	In vitro	Analysis of fluorescein passage in bovine brain micro vessel endothelial cells (BBMEC)	100 μM	NA	Increase in the passage of fluorescein in BBMEC.
Tunblad et al., 2003 [10]	In vivo	PBN is administered using micro dialysis probes to evaluate its influence on the passage of morphine to the CNS in rats	20 mg/kg e.f.b. 20 mg/kg/h i.f.	NA	Increase in morphine half-life by almost twice in rat brain.
Wei et al., 2015 [132]	In vivo	Cerebral ischemia/reperfusion (I/R) rat model	2 mg/kg i.p. 5 mg/kg gavage 0.1–1–10 mg/mL i.v.	Cerebral ischemia	Reduction in CA1 neuron loss and inflammation.
Yu X. Y. et al., 2007 [133]	Ex vivo	Administration of cytosine with PBN in rats and evaluation of its role	200 μM	AD	Decreases in cytosine output.
Zhang et al., 2019 [134]	In vivo	Sepsis-associated encephalopathy (SAE) mouse model	50 mg/kg i.p.	SAE	Attenuation in neuroinflammatory response and cognitive impairments.

Co = concentration. i.c.v. = intracerebroventricular. i.h. = intrahippocampal. i.p. = intraperitoneal. e.f.b. = bolus. i.f. = constant infusion. i.v. = intravenous. CNS = Central Nervous System. PD = Parkinson’s disease. AD = Alzheimer disease. NA = not applicable.

## Data Availability

Not applicable.

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
