# Peer review of "Probenecid, an Old Drug with Potential New Uses for Central Nervous System Disorders and Neuroinflammation"

_biomedicines, 2023, doi:10.3390/biomedicines11061516_

Round 1
Reviewer 1 Report
The manuscript "Probenecid, a potential pharmacological agent for the study of central nervous system disorders and neuroinflammation" by Claudia García-Rodriguez et al is aimed to review some recently discovered effects of probenecid in some conditions affecting the central nervous system (CNS) characterized by neuroinflammation such as epilepsy, multiple sclerosis, autoimmune encephalomyelitis, sepsis-associated encephalopathy, Alzheimer’s and Parkinson’s diseases.
Authors concluded that novel applications of probenecid is promising and presented studies using probenecid in both animal models and clinics. The authors have done a great job, the review looks accurate and informative and I have no objections to the essence of the manuscript. However, there are some questions.
The authors quite rightly mention gap junctions, their interaction with Probenecid and their role in the development of a number of pathologies, including epilepsy. This topic has been intensively studied in recent years and it was right to cite in the review some of the primary data published here:
Sahu et al, Pannexins form gap junctions with electrophysiological and pharmacological properties distinct from connexins; Scientific Reports v 4,; 4955 (2014)
Singh et al, Mechanisms of Spontaneous Electrical Activity in the Developing Cerebral Cortex-Mouse Subplate Zone; Cereb Cortex . 2019 ; 22;29(8):3363-3379
Volnova et al, The Anti-Epileptic Effects of Carbenoxolone In Vitro and In Vivo; Int J Mol Sci ; 2022 Jan 8;23(2):663. doi: 10.3390/ijms23020663
Bunse et al, The potassium channel subunit Kvbeta3 interacts with pannexin 1 and attenuates its sensitivity to changes in redox potentials; FEBS J . 2009;276(21):6258-70. doi: 10.1111/j.1742-4658.2009.07334.x.
Some researchers have drawn attention to some similarities between Probenecid and carbenoxolone, although the latter is not an antibiotic. Are there any opinions on this matter? I would suggest using this references above and adding a few lines of questions to the manuscript.
Minor criticisms:
In the Figure 2, in some places, a small light font is used on a light background. It seems to me that it would be right to somehow increase the contrast. Now it is quite difficult to see and read.
Except of these points, the paper is well organized and written clearly. I will be happy to recommend it for publication after corrections mentioned above.
Author Response
The manuscript "Probenecid, a potential pharmacological agent for the study of central nervous system disorders and neuroinflammation" by Claudia García-Rodriguez et al is aimed to review some recently discovered effects of probenecid in some conditions affecting the central nervous system (CNS) characterized by neuroinflammation such as epilepsy, multiple sclerosis, autoimmune encephalomyelitis, sepsis-associated encephalopathy, Alzheimer’s and Parkinson’s diseases.
Authors concluded that novel applications of probenecid is promising and presented studies using probenecid in both animal models and clinics. The authors have done a great job, the review looks accurate and informative and I have no objections to the essence of the manuscript. However, there are some questions.
Answer: Thanks to the reviewer for their positive reception of our manuscript. We also appreciate the thoughtful suggestions. As follows, we provide a detailed response to these comments.
The authors quite rightly mention gap junctions, their interaction with Probenecid and their role in the development of a number of pathologies, including epilepsy. This topic has been intensively studied in recent years and it was right to cite in the review some of the primary data published here.
Sahu et al, Pannexins form gap junctions with electrophysiological and pharmacological properties distinct from connexins; Scientific Reports v 4,; 4955 (2014)
Answer: This references are currently included in our manuscript.
Section 3.3, Page 6, line 247 and Section 4.5, page 14, line 542
Singh et al, Mechanisms of Spontaneous Electrical Activity in the Developing Cerebral Cortex-Mouse Subplate Zone; Cereb Cortex . 2019 ; 22;29(8):3363-3379
Answer: Section 4.5, page 14, line 545
Volnova et al, The Anti-Epileptic Effects of Carbenoxolone In Vitro and In Vivo; Int J Mol Sci ; 2022 Jan 8;23(2):663. doi: 10.3390/ijms23020663
Answer: Section 4.2, page 9, line 404
Bunse et al, The potassium channel subunit Kvbeta3 interacts with pannexin 1 and attenuates its sensitivity to changes in redox potentials; FEBS J . 2009;276(21):6258-70. doi: 10.1111/j.1742-4658.2009.07334.x.
Some researchers have drawn attention to some similarities between Probenecid and carbenoxolone, although the latter is not an antibiotic. Are there any opinions on this matter? I would suggest using the references above and adding a few lines of questions to the manuscript.
Answer: Suggestion accepted. We’ve included the suggested references in the new version of the manuscript and now we include a paragraph regarding the use of carbenoxolone in epilepsy and similitudes with the probenecid effects. Page 9, lines 401-408.
Minor criticisms:
In the Figure 2, in some places, a small light font is used on a light background. It seems to me that it would be right to somehow increase the contrast. Now it is quite difficult to see and read.
Except of these points, the paper is well-organized and written clearly. I will be happy to recommend it for publication after corrections mentioned above.
Answer: Thanks to the reviewer for their comments. We agree with the reviewer that Figure 2 is hard to read and see the details. We include a new version of Figure 2 in which we increased the font, image size, and resolution.

Reviewer 2 Report
This work reviews the Janus-like pharmacological features of Probenecid (PBN), as they can be either harmful or beneficial according to circumstances. Given the presently available evidence, the authors' optimism about PBN use in clinical settings against neurodegenerative diseases is a bit excessive. My comments are as follows:
1. The Abstract is nearly only an introduction to the PBN topic: in it there is neither a summary of the actual contents of the manuscript nor any conclusions. Actually, the Conclusions would be a better Abstract.
2. As the authors state "PBN blocks the active transport across the BBB": but are we sure that this is a real asset under any circumstance? Or is this feature more harmful than beneficial? It favours the intra-brain accumulation of potentially toxic agents, like neurotransmitters' metabolites, various xenobiotics, and morphine, for instance. Conversely, why hydroxyurea is not affected by PBN? These points need further clarifications and discussions. A clearer distinction is needed between preclinical advantages and effective clinical benefits or harms. Also it should be made clear that the positive results of animal brain studies usually fail to be translated to human brain studies.
3. Neuroinflammation is a very complex phenomenon being caused by multiple etiologic factors and sustained by different molecular mechanisms, which are not sufficiently discussed in this manuscript. For example only the NLRP3 and AIM2 inflammasome are marginally mentioned while omitting the well ascertained fact that there are many more inflammasomes in the animal and human brains and other organs and that hindering the activation of one or two inflammasomes does not obligatorily attenuates or switches off neuroinflammation.
4. The known multiple pharmacological interactions of PBN could also be the cause of worry particularly in aged patients, who often assume more than one drug.
5. PBN side effects are only listed, but they could be quite important for their specific prevalence, intensity, and collateral effects. A more detailed treatment of this topic is warranted if PBN is a candidate drug for neurodegenerative diseases due to the lengthy duration of the latter.
6. Several authors' serious statements cast doubt on the real relevance of a potential therapeutic use of PBN in neurodegenerative diseases, such as "is not entirely clear" and that in AD " the neurodegenerative hallmarks were not reduced by the acute PBN treatment" and "the (neuro)protective effects were not exclusively linked to the use of PBN" and "administered as a mono-drug for AD treatment it did not achieve sufficient benefits".
7. Figure 1 is very colorful but shows a P2XY7R (? may be P2X7R) and omits that its signaling driven by ATP hemichannel currents without altering P2X7 signaling. Does this double effect mean that in reality inflammasomes' activation is or is not affected? Or if P2X7 blockade is due to a PANX1-independent mechanism, could you mention the nature of the latter?
8. Figure 2 is also colorful but is difficult to read (too small characters) and somehow confusing.
9. As mentioned before, the Conclusions should more properly be the Abstract. They are also a bit too optimistic because they exclude the potentially harmful effects of PBN. Moreover, they do not mention clearly the manifold experimental models used to gain the related results. Finally, they should state clearly that based on the presently reported data, PBN at its best might act as a symptomatic agent in neurodegenerative diseases, as its role as a drug targeting their etiologic factors is as for now devoid of any evidence.
Author Response
Reviewer 2:
This work reviews the Janus-like pharmacological features of Probenecid (PBN), as they can be either harmful or beneficial according to circumstances. Given the presently available evidence, the authors' optimism about PBN use in clinical settings against neurodegenerative diseases is a bit excessive. My comments are as follows:
- The Abstract is nearly only an introduction to the PBN topic: in it there is neither a summary of the actual contents of the manuscript nor any conclusions. Actually, the Conclusions would be a better Abstract.
Answer: We thank the reviewer for the careful consideration of our manuscript. We agree that the abstract was not illustrative enough of the review. We amended this problem, which was rewritten in the new version of our manuscript.
- As the authors state "PBN blocks the active transport across the BBB": but are we sure that this is a real asset under any circumstance? Or is this feature more harmful than beneficial? It favours the intra-brain accumulation of potentially toxic agents, like neurotransmitters' metabolites, various xenobiotics, and morphine, for instance. Conversely, why hydroxyurea is not affected by PBN? These points need further clarifications and discussions. A clearer distinction is needed between preclinical advantages and effective clinical benefits or harms. Also it should be made clear that the positive results of animal brain studies usually fail to be translated to human brain studies.
Answer: We thank the reviewer for pointing this out, which requires further explanation. Pioneer studies demonstrated that probenecid blocks the transport of acid metabolites from the brain tissue to blood [1-3] and CSF to blood [4, 5]. Nevertheless, it should be noted that there are several transporter members of the SLC family and PBN only acts on some of the BBB transporters, so there could be others that continue to act, preventing the accumulation of these possible toxic substances in the brain. We comment on this issue in Section 1, page 2, lines 58-70.
Regarding the fact that the evidence obtained from animal studies not necessarily can be translated to human studies, we’ve mentioned this point in the Conclusions section.
- Neuroinflammation is a very complex phenomenon being caused by multiple etiologic factors and sustained by different molecular mechanisms, which are not sufficiently discussed in this manuscript. For example only the NLRP3 and AIM2 inflammasome are marginally mentioned while omitting the well ascertained fact that there are many more inflammasomes in the animal and human brains and other organs and that hindering the activation of one or two inflammasomes does not obligatorily attenuates or switches off neuroinflammation.
Answer: We agree with the reviewer that Neuroinflammation is a complex response that involves several molecular players and cellular pathways. However, we need to clarify that our interest was to write a review focused on the effects of PBN and not on the inflammasome types and activation mechanisms. Therefore, we decided not to deepen the different mechanisms and cascade of events. However, we added a couple of lines and references regarding the complex process of Neuroinflammation. (See Section 4.1, pages 7-8, lines 305-309 and 316-322), focusing the inflammatory pathways where pBN has been implicated.
Regarding the evidence obtained from animal studies, despite the evidence indicating a reduction in neuroinflammation under PBN treatment, we added a phrase that argues that the intervention of some inflammatory pathways does not imply a reduction in others. (See Section 5, page 15, lines 567-575)
- The known multiple pharmacological interactions of PBN could also be the cause of worry particularly in aged patients, who often assume more than one drug.
Answer: We thank the reviewer for pointing this out. We comment on this issue in Section 2, page 3, lines 100-105.
- PBN side effects are only listed, but they could be quite important for their specific prevalence, intensity, and collateral effects. A more detailed treatment of this topic is warranted if PBN is a candidate drug for neurodegenerative diseases due to the lengthy duration of the latter.
Answer: Although, in principle, we agree, a long time ago was already commented by other authors that these side effects or undesired symptoms subside when the dose is readjusted [6]. A long-lasting disease might indeed require long-term treatment, but this is just a possibility, and we do not know how long it could take to see effects. Since all is speculative, we mentioned the following in Section 2, page 3, lines 102-108: “There is no specific information comparing the use of probenecid in the elderly with its use in other ages, but it must be considered that older people generally take more medication, so it is critical to becoming aware whether PBN exhibit interactions with any of these drugs before recommending its use”…
- Several authors' serious statements cast doubt on the real relevance of a potential therapeutic use of PBN in neurodegenerative diseases, such as "is not entirely clear" and that in AD " the neurodegenerative hallmarks were not reduced by the acute PBN treatment" and "the (neuro)protective effects were not exclusively linked to the use of PBN" and "administered as a mono-drug for AD treatment it did not achieve sufficient benefits".
Answer: We appreciate the reviewer’s comment, and we agree with this point. The use of PBN as a therapeutic agent for AD is still not entirely explored since, in most of the exposed cases, it was used to increase the bioavailability in the CNS of other molecules, and in the particular case where it was administered as a mono drug for AD treatment, it did not achieve sufficient benefits in neuropathological markers due to the acute form of administration (in vitro treatment of brain slices) and therefore the evidence is still preliminar. However, Flores-Muñoz et al., demonstrated that PBN relieves synaptic symptoms of AD in an animal model. We comment on this issue briefly in Section 4.4, page 13, lines 465-469.
- Figure 1 is very colorful but shows a P2XY7R (? may be P2X7R) and omits that its signaling driven by ATP hemichannel currents without altering P2X7 signaling. Does this double effect mean that in reality inflammasomes' activation is or is not affected? Or if P2X7 blockade is due to a PANX1-independent mechanism, could you mention the nature of the latter?
Answer: We have modified Figure 1 to amend the typo. Regarding ATP and P2X7R signaling, as mentioned by the reviewer, it has been reported that the knockdown or the blockade of Panx1 eliminates the dye uptake that had been attributed to P2X7R without alteration of the ionic current or Ca2+ influx mediated by it, and more interestingly, the authors demonstrate that Panx1 is essential for the IL-1 release and caspase activation dependent on P2X7R activation [7]. We apologize if our original Figure 1 was confusing regarding that point, but the aim was to indicate the activity of each PBN target and how the PBN blockade can affect this activity. Now we include some modifications in Figure 1.
- Figure 2 is also colourful but is difficult to read (too small characters) and somehow confusing.
Answer: We have modified Figure 2 to make it easier to see and read.
- As mentioned before, the Conclusions should more properly be the Abstract. They are also a bit too optimistic because they exclude the potentially harmful effects of PBN. Moreover, they do not mention clearly the manifold experimental models used to gain the related results. Finally, they should state clearly that based on the presently reported data, PBN at its best might act as a symptomatic agent in neurodegenerative diseases, as its role as a drug targeting their etiologic factors is as for now devoid of any evidence.
Answer: Thanks for your comments. We've rewritten the abstract according to the reviewer's suggestions and reduced the strength of our claims in the conclusion section.
References
1 Neef NH, Tozer TN, Brodie BB. Application of seady-state kinetics to studies of the transfer of 5-hydroxyindoleacetic acid from brain to plasma. J Pharmacol Exp Ther 1967; 158: 214-8
2 Neff N, Tozer T, Brodie B. A specialized transport system to transfer 5-HIAA directly from brain to blood. Pharmacologist 1964; 6: 162
3 Werdinius B. Effect of probenecid on the level of homovanillic acid in the corpus striatum. J Pharm Pharmacol 1966; 18: 546-7
4 Olsson R, Roos BE. Concentrations of 5-hydroxyindoleacetic acid and homovanillic acid in the cerebrospinal fluid after treatment with probenecid in patients with Parkinson's disease. Nature 1968; 219: 502-3
5 Gottfires CG, Roos BE. Acid monoamine metabolites in cerebrospinal fluid from patients with presenile dementia (Alzheimer's disease). Acta Psychiatr Scand 1973; 49: 257-63
6 Boger WP, Strickland SC. Probenecid (benemid); its uses and side-effects in 2,502 patients. AMA Arch Intern Med 1955; 95: 83-92
7 Pelegrin P, Surprenant A. Pannexin-1 mediates large pore formation and interleukin-1beta release by the ATP-gated P2X7 receptor. Embo j 2006; 25: 5071-82

Round 2
Reviewer 2 Report
In Figure 2 words in yellow colour against a yellow or white background cannot be easily read
Author Response
Reviewer 2 (Round 2):
In Figure 2 words in yellow colour against a yellow or white background cannot be easily rea
Answer: We thank the reviewer for his comment regarding Figure 2. We agree that the light colour of the background inside the figure and the light colour font is hard to see. We amended this problem, as seen in the new version of Figure 3.
